# MERINO: ENTROPY-DRIVEN DESIGN FOR MOBILE-FRIENDLY GENERATIVE LANGUAGE MODELS

## ABSTRACT

Generative Large Language Models (LLMs) stand as a revolutionary advancement in the modern era of artificial intelligence (AI). However, deploying LLMs to resource-constrained devices is difficult due to their high computational cost. In this paper, we propose a novel information-entropy framework for designing mobile-friendly generative language models. Our key design paradigm is to maximize the entropy of transformer decoders within the given computational budgets. The whole design procedure involves solving a mathematical programming (MP) problem, which can be done on the CPU within minutes, making it nearly zero-cost. We evaluate our designed models, termed MeRino, across twelve NLP downstream tasks, showing their competitive performance against the state-of-the-art autoregressive transformer models under the mobile setting. Notably, MeRino achieves similar or better zero and one-shot performance compared to the 350M parameter OPT while being $4.9\times$ faster on mobile devices with $5.5\times$ reduction in model size.

## 1 INTRODUCTION

The Transformer architecture, originally introduced in (Vaswani et al., 2017), has revolutionized the field of natural language processing (NLP). It has become the de-facto building block in many large-scale pre-trained language models (LLMs) (Devlin et al., 2019; Liu et al., 2019; Radford et al., 2019; Brown et al., 2020; Zhang et al., 2022; Touvron et al., 2023). Especially, Generative Large-scale Language Models (LLMs), exemplified by GPT (Radford et al., 2019; Brown et al., 2020) and LLaMA (Touvron et al., 2023), have gained considerable popularity in recent studies. Yet, such models are without a doubt expensive to train and deploy. For instance, GPT-175B contains over 175 billion parameters, rendering it unsuitable for direct deployment on resource-constrained devices, such as mobile phones or Internet-of-Things (IoT) hardware. Consequently, there exists a substantial demand for developing lightweight language models that can be deployed to mobile systems with small memory footprints and low compute power.

A key challenge of designing mobile-friendly language models is that the hardware configuration varies from device to device. Therefore, it is difficult to design a one-fits-all model that satisfies all requirements. To this end, it is critical to customize an optimized language model backbone under different computational budgets. A conventional approach is to use knowledge distillation (KD) (Hinton et al., 2015) which distills larger language models into pre-defined smaller backbones (Li et al., 2021; Sanh et al., 2019; Sun et al., 2019). However, there is no guarantee that these pre-defined, fixed-size backbones are optimal on the given device. Another more flexible approach is to use AutoML (Hutter et al., 2019) or neural architecture search (NAS) (Wang et al., 2020; Xu et al., 2021; Yin et al., 2021) to optimize the transformer backbone. However, these methods are usually computationally demanding, which involves super-net (Cai et al., 2019; 2018) training or even brute-force grid search. Such processes often consume considerable GPU hours and leave large carbon footprints. Moreover, training super-nets is a non-trivial task as child architectures often interfere with each other which leads to performance degradation, as reported in (Ning et al., 2020).

In this paper, we present an entropy-driven framework to design lightweight variants of generative language models tailored for resource-constrained devices. Our method leverages recent advancements in information theory and theoretical deep learning which formulate autoregressive language

models as information processing systems parameterized by structural parameters such as network widths and depths. Then, the Maximum Entropy Principle (Jaynes, 1957) is applied to optimize the network architecture design. More specifically, our design aims to find the optimal configuration of network structure parameters, including depths/widths/embedding dimensions, such that the network entropy is maximized under the given computational budgets, such as parameter size and FLOPs.

Albeit the Maximum Entropy Principle is conceptually simple, a direct application encounters two technical challenges. First, the notion of entropy for a transformer backbone is not well-defined in deep learning literature. To overcome this hurdle, we propose to use subspace entropy spanned by the network parameters at random initialization as model entropy. The computation of subspace entropy can be accelerated via table lookup therefore is highly efficient. Second, we find that naively maximizing the entropy will lead to an over-deep transformer backbone that is difficult to train. To address this issue, we propose to preserve the model trainability during the architecture design. Then an Evolutionary Algorithm (EA) is utilized to optimize the structural parameters of the transformer backbone (e.g., number of heads, channels, embedding dimensions, etc.). Finally, we are able to design a family of optimized, **M**obile-fri**e**ndly gene**R**ative la**n**guage m**o**dels, or **MeRino** for short, on various mobile devices at nearly zero cost.

The key contributions of this work are summarized as follows:

- To the best of our knowledge, we first present an entropy-driven framework to address the challenge of designing efficient generative language models for resource-constrained devices at nearly zero cost.
- Our framework leverages the Maximum Entropy Principle and considers both the entropy and trainability of language models to optimize transformer architectures given computation budgets.
- Experimental results show that MeRino achieve competitive performance against the state-of-the-art LLMs, including OPT and GPT models. Notably, our models exhibit improved parameter, computation, and throughput efficiency on mobile devices.

## 2 RELATED WORK

**Generative Large Language Models (LLMs)** Generative large language models (LLMs) have emerged as the standard solution to a wide range of NLP tasks. They are generally pre-trained on large-scale corpora in self-supervised manners to learn the contextual structure of natural language. Unlike previous language models, LLMs consist of only transformer decoder layers and exhibit outstanding ability to scale up and impressive zero-shot generalization performances. GPT-3 (Brown et al., 2020), in particular, pushed the boundaries of casual language models by scaling up the model size to 175 billion parameters and pre-training on a large corpus of over 570 GB plain texts. In the pursuit of democratizing and fostering reproducible research in LLMs, Meta AI recently released Open Pre-trained Transformers (OPT) (Zhang et al., 2022), a suite of decoder-only models, ranging from 125 M to 175 B parameters. In this work, our scope is generative, or decoder-only transformer-based language models and we aim to design such LLMs suitable for mobile devices with limited memory space and compute power.

**Knowledge Distillation (KD)** One of the most widely studied techniques in compressing LLMs is knowledge distillation (KD) (Hinton et al., 2015). BERT-PKD (Sun et al., 2019) distill BERT into smaller students using knowledge transfer in both final output and hidden states in multiple intermediate layers. TinyBERT (Jiao et al., 2019) adopts a layer-wise distillation strategy for BERT at both the pre-training and fine-tuning stages. (Li et al., 2021) investigates numerous KD techniques to compress GPT-2 models by layer truncation. Despite achieving promising results, the above KD-based methods can only distill LLMs into a fixed-size model, which is not suitable for deployment on diverse and heterogeneous devices. In this work, orthogonal to KD, which focuses primarily on the training and fine-tuning stage, our proposed method emphasizes designing lightweight transformer architectures with various parameter sizes and FLOPs to meet different hardware constraints.

**NAS for NLP** Due to its success in computer vision (CV), neural architecture search (NAS) has recently gained attention in the NLP community. NAS-BERT (Xu et al., 2021) trains a supernet to

efficiently search for masked language models which are compressed versions of the standard BERT. AutoTinyBERT (Yin et al., 2021) further reduces overall computation cost over NAS-BERT by adopting a linear search space. For encoder-decoder architectures, HAT (Wang et al., 2020) uses the Once-For-All (Cai et al., 2019) approach and performs a search on sub-samples of the supernet that inherits weights to estimate downstream task accuracy. LTS (Javaheripi et al., 2022) proposes using non-embedding parameters in decoders as a proxy score to predict the perplexity performance of generative LLMs. However, the aforementioned methods are mostly data-dependent and incur heavy computation costs. Moreover, it is difficult for researchers to understand why specific architectures are preferred by the algorithm and what theoretical insight we can learn from these results. In this work, we plan to explore the architecture design of autoregressive language models in a principled way with clear theoretical motivation and human explainability.

**Information Theory in Deep Learning**   Information theory recently has emerged as a powerful tool for studying deep neural networks (Chan et al., 2021; Saxe et al., 2018; Shen et al., 2023; Sun et al., 2021). Several previous studies (Chan et al., 2021; Saxe et al., 2018) have attempted to establish a connection between the information entropy and the neural network architectures. For instance, (Chan et al., 2021) tries to interpret the learning ability of deep neural networks using subspace entropy reduction. (Saxe et al., 2018) investigates the information bottleneck in deep architectures and explores the entropy distribution and information flow in deep neural networks. Additionally, (Shen et al., 2023; Sun et al., 2021) focus on designing high-performance convolutional neural networks (CNNs) via maximizing multi-level entropy. Yet, to the best of our knowledge, there is still no published work using information entropy to design efficient decoder-only transformer backbones for language models.

## 3 METHODOLOGY

In this section, we begin by presenting some preliminary details on autoregressive transformer models. Next, we introduce our novel definition of network entropy for transformer models. Moreover, we demonstrate that the untrained subspace entropy positively correlates with the model performance after training. Finally, we present our entropy-driven design procedure, which solves a constrained mathematical programming problem using the Evolutionary Algorithm (EA).

### 3.1 PRELIMINARIES

Due to the page limit, we present preliminary details of autoregressive transformer models in **Appendix A**. For notation purposes, we denote the attention matrices in MHA as $W^Q, W^K, W^V \in \mathbb{R}^{d \times d/h}$ for queries $Q$, keys $K$, and values $V$, respectively and output project matrix as $W^O \in \mathbb{R}^{d \times d}$; for FFN layers, we denote two linear project matrices as $W^{\text{FFN}_1} \in \mathbb{R}^{d \times rd}, W^{\text{FFN}_2} \in \mathbb{R}^{rd \times d}$, where $r$ is the FFN ratio.

### 3.2 SUBSPACE ENTROPY FOR TRANSFORMERS

**Expressiveness in Deep Network**   From the perspective of information theory (Jaynes, 1957; Cover & Thomas, 1991), deep neural networks can be regarded as information systems, and their performance is closely related to the expressive power of such networks. The notion of entropy is often used to measure such expressiveness through intermediate feature maps (Sun et al., 2021) in convolutional neural networks (CNNs). In the case of transformers, we propose to define the entropy of transformers from the perspective of parameter subspaces.

Suppose that $W_i \in \mathbb{R}^{c_{\text{in}}^{(i)} \times c_{\text{out}}^{(i)}}$ presents a linear mapping with $c_{\text{in}}^{(i)}$ input channels and $c_{\text{out}}^{(i)}$ output channels. The elements of $W_i$ are randomly sampled from the standard Gaussian distribution $\mathcal{N}(0, 1)$. According to previous works (Chan et al., 2021), the subspace spanned by the random linear mapping $W_i$ has entropy defined by

$$\widehat{H}(W_i) \triangleq \mathbb{E}\{\sum_{j=1}^{r_i} \log(1 + \frac{s_j^2}{\epsilon^2})\} \tag{1}$$

where $r_i = \min(c_{\text{in}}, c_{\text{out}})$, $s_j$ is the $j$-th largest singular value of $W_i$ and $\epsilon$ is a small constant.

For an $L$-layer network $f(\cdot)$, we define the network entropy $\widehat{H}(f)$ by accumulating the entropy of matrices in each layer as the following:

$$\widehat{H}(f) = \sum_{i=1}^{L} \widehat{H}(W_i) \tag{2}$$

**Effectiveness in Deep Network**     The entropy measures the *expressiveness* of the deep neural network, which is positively correlated with the network performance (Sun et al., 2021). However, directly maximizing the above-defined entropy leads to the creation of over-deep networks, since according to Eq. (2), the expressivity (entropy) grows exponentially faster in depth (number of layers $L$), than in width (dimension of $W_i$). For an over-deep network, a small perturbation in low-level layers of the network will lead to an exponentially large perturbation in the high-level output of the network (Roberts et al., 2021). During the back-propagation process, the gradient flow often cannot effectively propagate through the entire network. Though recent works have attempted to alleviate the trainability issues by revising initialization strategies (Zhang et al., 2019; Huang et al., 2020), adding skip connections (Nguyen & Salazar, 2019; He et al., 2015), or proposing better architectures (Wang et al., 2019b; Bachlechner et al., 2020), training over-deep networks still remains a rather challenging problem.

Table 1: Perplexity comparison of two different structures of autoregressive transformer models on the LM1B dataset.

| Model | $L$ | $E$ | Params | Entropy | Effective $\gamma$ | Entropy w/ $\gamma$ | Validation PPL |
|---|---|---|---|---|---|---|---|
| 'Wide' | 1 | 256 | 40 M | 2784 | **0.008** | **2243** | **53.7** |
| 'Deep' | 24 | 64 | 40 M | **4680** | 0.25 | 2042 | 71.9 |

To verify the negative impact when the network is over-deep, in Table 1, we conduct experiments of training two transformer architectures with a similar parameter size of 40 M. One model, referred to as the 'Wide' model, consists of only one layer and an embedding dimension of 256. The other model, referred to as the 'Deep' model, consists of 24 layers but only with an embedding dimension of 64. Both models are trained under the same setting until convergence. We observe that even though the 'deep' network has much higher entropy, it obtains worse perplexity performance after training than the 'wide' network. This observation aligns with the common belief that over-deep networks hinder effective information propagation (Roberts et al., 2021) and are difficult to train and optimize (Rae et al., 2021).

To address the potential trainability issues, we propose adding additional constraints to control the depth-width ratio of networks. Specifically, we adopt the term *effectiveness* $\gamma$ from the work (Roberts et al., 2021) and define it as follows:

$$\gamma = \beta L / \hat{w} \tag{3}$$

Here, $\hat{w}$ is the effective width of a $L$-layer network and $\beta$ is a scaling factor to control $\gamma$ within the range of 0 and 1. To enforce the above constraint, we revise Eq. (2) as follows:

$$\widehat{H}(f) = (1 - \gamma) \sum_{i=1}^{L} H(W_i) \tag{4}$$

Compared to the previous subspace entropy definition, Eq. (4) penalizes networks with larger depth-to-width ratios (higher $\gamma$). This constraint helps alleviate potential trainability issues by promoting a more balanced depth-width ratio in the network architecture. By considering both *expressiveness* (entropy) and *effectiveness* (the depth-width ratio), we aim to design more capable and trainable models.

**Entropy of Transformers**     Consider a $L$-layer transformer model with embedding dimension $E$ and FFN dimension $F$, according to Theorem 1 in (Levine et al., 2020), the depth-width sufficiency behavior satisfied a logarithmic condition in transformer models. Subsequently, we propose to define

the effective width of MHA and FFN and their corresponding entropy as:

$$\hat{w}_{\text{MHA}} = \log E, \qquad \hat{w}_{\text{FFN}} = \log F \tag{5}$$

$$\widehat{H}_{\text{MHA}} = (1 - \frac{\beta L}{\hat{w}_{\text{MHA}}}) \sum_{i=1}^{L} \widehat{H}(W_i^Q, W_i^K, W_i^V, W_i^O) \tag{6}$$

$$\widehat{H}_{\text{FFN}} = (1 - \frac{\beta L}{\hat{w}_{\text{FFN}}}) \sum_{i=1}^{L} \widehat{H}(W_i^{\text{FFN}_1}, W_i^{\text{FFN}_2}) \tag{7}$$

In practice, we find that using weighted entropy for MHA and FFN gives us a more reliable indicator for model performance. Therefore, we define the total entropy of the transformer model as linear combinations of the MHA and FFN entropy:

$$\widehat{H} = \alpha_1 \widehat{H}_{\text{MHA}} + \alpha_2 \widehat{H}_{\text{FFN}} \tag{8}$$

where $\alpha = (\alpha_1, \alpha_2)$ are tunable hyperparameters.

**Fast Entropy Approximation**     Given the above definitions, we can easily calculate entropy for any transformer model. However, performing singular value decomposition (SVD) is a costly operation. For large models, it sometimes requires minutes to run SVD, which inhibits a zero-cost design. To accelerate the entropy computation, we build an entropy lookup table to approximate the total entropy of a given transformer model. The lookup table is built through a pre-computation process that considers all possible combinations of expected entropy values for different dimensions. This step incurs only a one-time cost and the resulting lookup table can be shared across multiple experiments. With the lookup table in place, we can efficiently calculate the entropy of transformer models and enable a more efficient design process for transformer models.

**Evaluating Transformer without Training**     Recent studies (Jaynes, 1957; Shen et al., 2023) have demonstrated that entropy, which captures the information capacity of neural network architecture, can be a reliable indicator for performance and generalization ability (Jaynes, 1957; Shen et al., 2023) in convolutional neural networks. In this part, we provide experimental results that empirically establish a strong correlation between our proposed entropy of untrained transformers and their final performance on the LM1B (Chelba et al., 2013) dataset after training. Figure 2 illustrates the correlation between the model performance (negative perplexity) and their corresponding entropy scores. Results indicate strong correlations, as evidenced by Spearman's Rank Correlation ($\rho$) and Kendall Rank Correlation ($\tau$) scores exceeding 0.8 and 0.6, respectively. This suggests that entropy can serve as a reliable training-free proxy for evaluating transformer architecture.

We recognize that while our method approach has some connections to zero-shot NAS (Lin et al., 2021; Sun et al., 2021; Zhou et al., 2022), there are two principal distinctions. First, zero-shot NAS methods are predominantly *data-driven*. Our method, on the other hand, is mathematically driven with clear motivation from the perspective of information theory. Second, zero-shot NAS methods are inherently *data-dependent*, requiring forward and backward passes over the architecture. Such processes often need to store network parameters and feature maps in GPU memory. In contrast, our methodology is purely analytical and the expensive entropy calculation process is substituted by table lookup procedure, therefore highly efficient, and truly zero-cost. **Our method requires zero GPU memory and zero GPU core in the design stage**. In summary, our method is a much better approach to designing efficient language models for mobile devices than zero-shot NAS.

### 3.3 DESIGNING MOBILE LANGUAGE MODELS

**Search Space**     In the design of MeRino, we introduce an adaptive block-wise search space to construct the backbone architecture. This allows us to determine the architectural parameters on a per-block basis. Each transformer block consists of numerous transformer layers of the same number of attention heads, hidden dimensions, and embedding dimensions. Within each transformer block, in MHA layers, we fix the head dimension and make the attention head number elastic so that each attention module can decide its necessary number of heads. We also set the *Q-K-V* dimensions the same as embedding dimensions; in FFN layers, the hidden dimension is decided by choosing the FFN ratio to the embedding dimension. To prevent information bottlenecks, we also ensure that as the network goes deeper, the embedding dimension of each transformer block should be

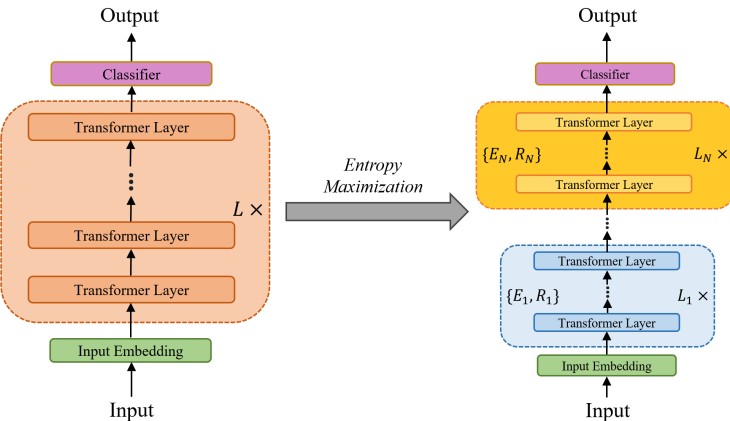

Figure 1: Our proposed adaptive block-wise transformer design. Left is the standard autoregressive transformer design, which consists of $L$ homogeneous layers, and right is the optimal architecture design after entropy maximization, where there are $N$ number of transformer blocks and each transformer block has adaptive width $(E_i, R_i)$ and depth $(L_i)$.

non-decreasing. Moreover, we incorporate parameter sharing technique (Lan et al., 2019) within each transformer block. This means that all MHA and FFN layers within the block share the same weights, resulting in transformer models of reduced memory footprint. Illustration can be found in Figure 1. Details of our search space configuration are provided in **Appendix C.1**.

**Search Process**    To design a transformer model $f(\cdot)$ with $N$ transformer blocks under a given computation budget $\mathcal{C}$, we propose to optimize the parameters $\{E_j, R_j, L_j\}_{j=1,...,N}$ by solving a mathematical programming (MP) problem. The objective of the MP problem is to maximize a weighted sum of entropy, representing the expressiveness and effectiveness of the model, while considering constraints on the computational cost. The MP problem is formulated as follows:

$$\max_{E_i, R_i, L_i} \quad \alpha_1 \sum_{j=1}^{N} L_j (1 - \frac{\beta L_j}{\log E_j}) \widehat{H}(E_j) + \alpha_2 \sum_{j=1}^{N} L_j (1 - \frac{\beta L_j}{\log(R_j E_j)}) \widehat{H}(R_j E_j) \tag{9}$$
$$\text{s.t.} \quad \text{ComputeCost}[f(\cdot)] \leq \mathcal{C}, \quad E_1 \leq E_2 \leq \cdots \leq E_N$$

where $E_j$, $R_j$, and $L_j$ denote the embedding dimension, FFN ratio, and number of layers in the $j$-th transformer block, respectively. To solve this optimization problem, we employ an **Evolutionary Algorithm** (Reeves, 2007). Note that Eq. (9) can be solved by any non-linear programming solver in principle. We choose EA due to its simplicity. Since our formulated problem is purely mathematical, it can be solved nearly instantly on the CPU. A detailed description of EA and the mutation algorithm is given in **Appendix C.3**.

## 4    EXPERIMENTS

In this section, we first describe experimental settings for search, training, and evaluation. Next, we report the results of MeRino on various NLP tasks and compare our approach with both existing pretrained LLMs and zero-shot NAS methods. Finally, we conduct ablation studies of different key components in MeRino.

### 4.1    EXPERIMENTAL SETTINGS

**Search Settings**    In searching for MeRino, the number of iterations $T$ is set to 100000, with a population size $M$ of 512 and the parent size $K$ of 64. We conduct searches for three different FLOP targets (60/110/160 G). We limit the number of transformer blocks to $N = 4$ and set $\alpha = (0.6, 0.4)$ and $\beta = 1/16$.

**Training Settings**    We mostly follow settings in (Zhang et al., 2022) and (Biderman et al., 2023) and pre-train our models on the Pile dataset (Gao et al., 2020) for 600k steps ($\approx$ 314B tokens) with

an effective batch size of 512 using AdamW optimizer (Loshchilov & Hutter, 2017), with a starting learning rate of 6e-4 and warm-up steps of 1000, and linear learning rate decay schedule. We also enable automatic mixed precision (AMP) for better training efficiency.

**Evaluation Settings**     We evaluate our models for zero and one-shot natural language inference tasks across twelve different downstream NLP tasks, namely HellaSwag (Zellers et al., 2019), WinoGrande (Sakaguchi et al., 2019), OpenBookQA (Mihaylov et al., 2018), ARC-easy, and ARC-challenge (Clark et al., 2018), PubmedQA (Jin et al., 2019), LogiQA (Liu et al., 2020), and Super-GLUE (Wang et al., 2019a) benchmark BoolQ, CB, WIC, WSC and RTE. FLOPs are calculated with a batch size of 1 and sequence length of 1024 and inference throughput is measured at token per second on NVIDIA Jetson Nano 8GB.

Table 2: Detailed zero-shot downstream task results for MeRino and publicly available pretrained LLMs.

| | MeRino | | | OPT | | Pythia | | Cerebras-GPT | GPT-2 |
|---|---|---|---|---|---|---|---|---|---|
| Params (↓) | 52 M | 61 M | 64 M | 125 M | 350 M | 70 M | 162 M | 111 M | 124 M |
| FLOPs (↓) | 60 G | 110 G | 160 G | 210 G | 720 G | 100 G | 270 G | 260 G | 290 G |
| Throughput (↑) | 36.37 | 33.85 | 25.97 | 23.84 | 6.38 | 27.25 | 14.03 | 22.49 | 19.06 |
| HellaSwag | 0.267 | 0.273 | 0.274 | 0.267 | 0.283 | 0.269 | 0.292 | 0.267 | **0.300** |
| WinoGrande | 0.507 | 0.510 | 0.528 | 0.503 | 0.523 | **0.529** | 0.492 | 0.490 | 0.516 |
| ARC-Easy | 0.327 | 0.336 | 0.341 | 0.386 | **0.389** | 0.335 | 0.373 | 0.336 | 0.382 |
| ARC-Challenge | 0.212 | 0.209 | **0.234** | 0.223 | 0.233 | 0.214 | 0.231 | 0.207 | 0.230 |
| OpenBookQA | 0.242 | 0.248 | 0.267 | 0.226 | **0.286** | 0.272 | 0.264 | 0.256 | 0.272 |
| BoolQ | 0.541 | 0.610 | **0.621** | 0.554 | 0.618 | 0.589 | 0.571 | 0.621 | 0.554 |
| WIC | **0.525** | 0.502 | 0.505 | 0.498 | 0.500 | 0.486 | 0.500 | 0.500 | 0.492 |
| CB | 0.411 | 0.375 | 0.393 | 0.357 | **0.464** | 0.339 | 0.446 | 0.411 | 0.410 |
| WSC | 0.413 | 0.365 | 0.375 | 0.365 | 0.365 | 0.365 | 0.365 | 0.365 | **0.433** |
| RTE | 0.502 | 0.534 | 0.545 | 0.444 | 0.542 | 0.523 | **0.563** | 0.549 | 0.531 |
| PubmedQA | 0.377 | 0.484 | 0.540 | 0.372 | 0.414 | 0.409 | 0.544 | **0.552** | 0.430 |
| LogiQA | 0.276 | 0.255 | 0.278 | **0.286** | 0.280 | 0.266 | 0.269 | 0.266 | 0.245 |
| Average | 0.383 | 0.392 | 0.408 | 0.373 | 0.408 | 0.383 | **0.409** | 0.402 | 0.400 |

## 4.2 MAIN RESULTS

**Comparison with Pre-trained LLMs**     Since our scope is mobile-friendly language models, we mainly compare pretrained LLMs that can be run on NVIDIA Jetson Nano 8GB with out-of-memory (OOM) issues. We compare the average accuracy of our MeRino models with baseline models, such as GPT-2 (Radford et al., 2019), OPT (Zhang et al., 2022), Pythia (Biderman et al., 2023) and Cerebras-GPT (Dey et al., 2023).

Table 2 reports the comparisons of MeRino and current state-of-the-art autoregressive transformer-based language models. Compared to the OPT family, MeRino achieves superior accuracy with much less parameter size and FLOPs. Specifically, MeRino-64M obtains similar average accuracy as OPT-350M but with 82% and 78% reduction in model size and computation respectively. For similar inference throughput performance, MeRino-64M outperforms OPT-125M by 3.5%. Above all, MeRino achieves an average inference speedup of 2.7× against OPT family models, respectively.

When compared to open-sourced LLMs that are trained on the Pile dataset, MeRino-64M achieves 0.6% higher average zero-shot accuracy than Cerebras-GPT while reducing parameter size and FLOPs by 1.7× and 1.6×, respectively; MeRino-61M is also 0.8% more accurate than GPT-2 with 1.4× lower latency; our smallest model, MeRino-52M achieves similar performance as Pythia-70M but with 1.5× faster runtime. Similar trends can be found in the one-shot performance comparison results in **Appendix B**.

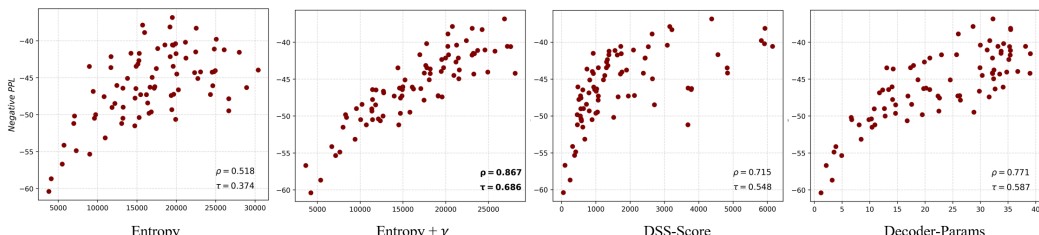

Figure 2: Correlation comparison of training-free predictor and transformer performance (negative perplexity). $\rho$ is Spearman's Rank and $\tau$ is Kendall Tau. Larger values mean higher correlation.

**Comparison with Zero-shot NAS**     We also compare our methods against two zero-shot NAS approaches, namely DSS-Score (Zhou et al., 2022) and Decoder-Params (Javaheripi et al., 2022). For correlation performance, we randomly and uniformly sample 81 unique transformer architectures in the standard autoregressive transformer search space. Each model is fully trained from *scratch* on the One Billion Word (LM1B) Chelba et al. (2013) dataset and the performance is measured using validation perplexity. According to the results in Figure 2, we can see that our proposed subspace entropy is more positively correlated with the final model perplexity performance than the other two training-free metrics.

Additionally, we conduct searches using the same FLOPs constraints (160 G), and report the downstream NLP performance of searched architectures at different iterations (0, 12k, 24k, 36k, 48k, 64k). In Figure 3, we can see that under the same computation constraint, our entropy-driven design can produce much more capable language models.

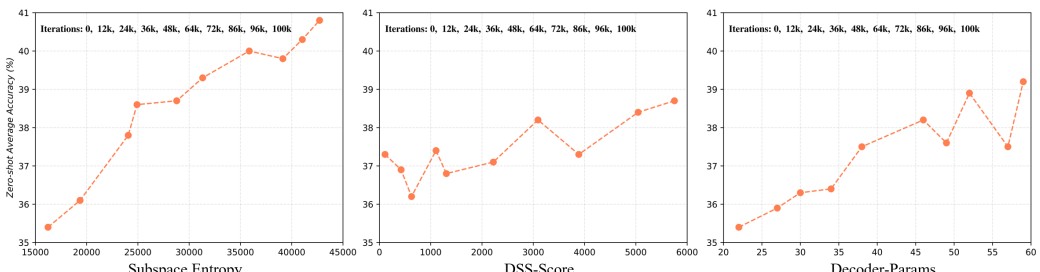

Figure 3: Avg zero-shot accuracy vs. different training-free proxies during searches. The dotted line indicates the evolution direction of the search process.

## 4.3 ABLATION STUDIES

**Effectiveness Constraint**     As shown in Table 3, effectiveness constraint $\gamma$ plays a key role in helping our entropy-driven framework design more capable and trainable models. When using effectiveness constraint $\gamma$, the final searched language model obtains +2.4% average accuracy gain. In terms of correlation experiments on the LM1B dataset shown in Figure 2, entropy with effectiveness constraint $\gamma$ can provide a more reliable prediction of the final perplexity performance of trained transformer models, especially in identifying high-performance architectures.

Table 3: Performance comparison of effectiveness constraint and weighted entropy. Inference throughput is measured on NVIDIA Jetson Nano 8GB.

| Model | Effectiveness Constraint | Weighted Entropy | Params (M) | FLOPs (G) | Throughput (token/s) | Avg. Zero-shot Accuracy |
|---|---|---|---|---|---|---|
| | ✗ | ✗ | 62 | | 33.27 | 0.360 |
| MeRino | ✓ | ✗ | 59 | 110 | 37.42 | 0.384 |
| | ✓ | ✓ | 61 | | 33.85 | **0.392** |

**Weighted Entropy**    We also study the impact of weight $\alpha$ on our entropy-driven approach. As shown in Figure 4, naively adding MHA and FFN without weights cannot represent the perplexity performance very well. Weighted entropy, on the other hand, especially when properly tuned, exhibits much better correlation results than unweighted entropy. In Table 3, we further evaluate the impact of weighted entropy on downstream performance. We can see that using weighted entropy helps improve the average zero-shot accuracy by 0.8%.

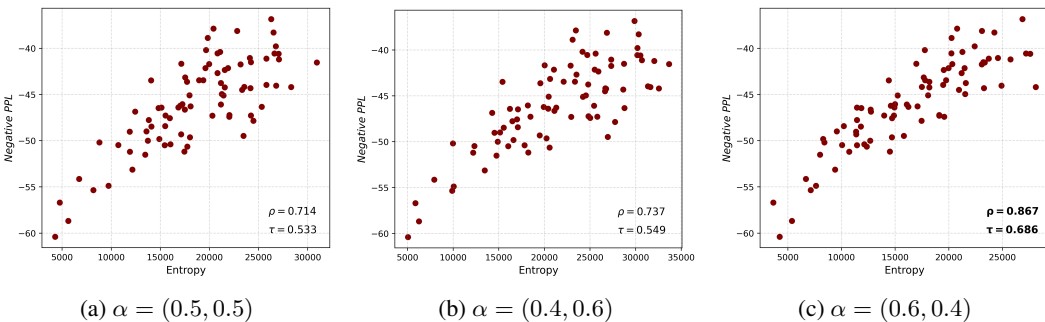

(a) $\alpha = (0.5, 0.5)$             (b) $\alpha = (0.4, 0.6)$             (c) $\alpha = (0.6, 0.4)$

Figure 4: Correlation results of different weighted entropy scores on LM1B dataset. $\rho$ is Spearman's Rank and $\tau$ is Kendall Tau.

**Parameter Sharing**    We report the effect of parameter technique on MeRino in Table 4 for three different FLOPs targets (60/110/160 G). We can see that sharing parameters within the same transformer block helps improve parameter efficiency and reduce the model size while having a negligible impact on both the language modeling (see Pile test loss) and downstream zero and one-shot performance.

Table 4: Performance comparison of parameter sharing technique under three different FLOPs target.

| Parameter Sharing | Params (M) | FLOPs (G) | Pile Test Loss | Downstream task performance | |
|:---:|:---:|:---:|:---:|:---:|:---:|
| | | | | Zero-shot | One-shot |
| | 59 | 60 | 2.496 | 0.381 | 0.382 |
| ✓ | 52 | | 2.520 | 0.383 | 0.387 |
| | 79 | 110 | 2.492 | 0.395 | 0.390 |
| ✓ | 61 | | 2.517 | 0.392 | 0.394 |
| | 100 | 160 | 2.378 | 0.403 | 0.402 |
| ✓ | 64 | | 2.381 | 0.408 | 0.403 |

## 5    CONCLUSION

In this paper, we present MeRino, a novel design framework aiming to generate efficient autoregressive language models for mobile devices, such as NVIDIA Jetson Nano. By modeling transformer models as information processing systems, MeRino leverages the Maximum Entropy Principle and optimizes the network architecture by maximizing the subspace entropy of transformer decoders and model trainability under given computational budgets. We show that MeRino can achieve comparable performance against state-of-the-art LLMs with significant improvement in model size reduction and inference runtime speedup on resource-constrained devices.

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

APPENDIX

In the appendix, we provide preliminary knowledge of autoregressive transformers (Appendix A), detailed one-shot learning results (Appendix B), design details of MeRino (Appendix C), including search space configurations of our entropy-driven design, structural details of MeRino, and detailed Evolutionary Algorithm (EA) and Mutation algorithm, and limitations (Appendix D).

## A  PRELIMINARIES

**Autoregressive Transformers**  Decoder-only, or autoregressive transformers, operate by predicting the next element in a sequence based on the preceding elements. A standard autoregressive transformer comprises an embedding layer to project sequences of tokens to hidden dimensions and stacks of transformer layers to capture long-term dependencies between input tokens using the self-attention mechanism. A transformer layer includes two main components: a multi-head attention (MHA) module and a position-wise feed-forward network (FFN). The MHA module facilitates capturing contextual information by attending to different positions within the input sequence, while the FFN performs element-wise transformations to introduce non-linearity and improve representational capacity.

**Multi-Head Attention (MHA)**  Multi-head attention (MHA) is a crucial component within the transformer architecture that enables the model to selectively attend to different segments of the input sequence. This mechanism involves projecting the input sequence into multiple attention heads, each of which calculates an independent attention distribution. In MHA computation, there are specifically four main matrices involved: attention matrices $W^Q, W^K, W^V \in \mathbb{R}^{d_{in} \times d_{in}/h}$ and output project matrix $W^O \in \mathbb{R}^{d_{in} \times d_{out}}$. Given the output of previous layers $X \in \mathbb{R}^{n \times d_{in}}$ as input, the attention function is formulated as:

$$Q, K, V = XW^Q, XW^K, XW^V \tag{10}$$

$$\text{Attn}(Q, K, V) = \text{softmax}(\frac{QK^T}{\sqrt{d_{in}/h}})(V) \tag{11}$$

where $Q$, $K$, and $V$ represent queries, keys, and values, respectively.

MHA is defined by concatenating $h$ attention heads and producing outputs as follows:

$$\text{MHA}(X) = \text{Concat}(\text{Attn}_i, ..., \text{Attn}_h)W^O \tag{12}$$

In addition, the transformer layer adopts residual connection and layer normalization on top of MHA to compute the final outputs.

$$X^{\text{MHA}} = \text{LayerNorm}(X + \text{MHA}(X)) \tag{13}$$

**Position-wise Feed-forward Network (FFN)**  In addition to the MHA, each transformer layer includes a feed-forward network (FFN). The FFN applies two point-wise fully connected layers followed by a non-linear activation function, such as ReLU. Operations within FFN can be formulated as follows:

$$X^{\text{FFN}} = \text{ReLU}(X^{\text{MHA}}W^{\text{FFN}_1} + b_1)W^{\text{FFN}_2} + b_2 \tag{14}$$

Similarly, the FFN also incorporates residual connections and layer normalization to compute the final outputs:

$$X^{\text{FFN}} = \text{LayerNorm}(X^{\text{MHA}} + X^{\text{FFN}}) \tag{15}$$

## B  ONE-SHOT LEARNING RESULTS

We report additional one-shot comparison results in Table 5. We can see that our designed models still achieve competitive performance against state-of-the-art LLMs with reduced parameters and computation.

Table 5: Detailed one-shot downstream task results for MeRino and publicly available pretrained LLMs.

| | MeRino | | | OPT | | Pythia | | Cerebras-GPT | GPT-2 |
|---|---|---|---|---|---|---|---|---|---|
| Params (↓) | 52 M | 61 M | 64 M | 125 M | 350 M | 70 M | 162 M | 111 M | 124 M |
| FLOPs (↓) | 60 G | 110 G | 160 G | 210 G | 720 G | 100 G | 270 G | 260 G | 290 G |
| HellaSwag | 0.262 | 0.260 | 0.270 | 0.264 | 0.279 | 0.266 | 0.296 | 0.265 | **0.308** |
| WinoGrande | 0.517 | 0.486 | 0.495 | 0.504 | 0.519 | **0.522** | 0.506 | 0.494 | 0.500 |
| ARC-Easy | 0.339 | 0.351 | 0.353 | 0.396 | **0.413** | 0.344 | 0.387 | 0.348 | 0.399 |
| ARC-Challenge | 0.214 | 0.208 | 0.237 | 0.229 | 0.238 | 0.208 | 0.225 | 0.218 | 0.235 |
| OpenBookQA | 0.234 | 0.240 | 0.262 | 0.232 | 0.258 | 0.238 | 0.266 | 0.262 | **0.266** |
| BoolQ | 0.536 | 0.539 | 0.570 | 0.547 | 0.583 | 0.521 | 0.560 | **0.605** | 0.526 |
| WIC | 0.467 | 0.489 | 0.472 | 0.483 | **0.506** | 0.464 | 0.467 | 0.475 | 0.464 |
| CB | 0.411 | 0.482 | **0.482** | 0.464 | 0.429 | 0.464 | 0.482 | 0.464 | 0.482 |
| WSC | **0.423** | 0.413 | 0.365 | 0.365 | 0.365 | 0.365 | 0.365 | 0.365 | 0.365 |
| RTE | **0.574** | 0.542 | 0.549 | 0.484 | 0.523 | 0.538 | 0.520 | 0.552 | 0.549 |
| PubmedQA | 0.404 | 0.466 | 0.513 | 0.444 | 0.462 | 0.478 | 0.521 | 0.463 | 0.425 |
| LogiQA | 0.264 | 0.256 | 0.269 | 0.246 | 0.252 | **0.284** | 0.258 | 0.255 | 0.250 |
| Average | 0.387 | 0.394 | 0.403 | 0.388 | 0.402 | 0.391 | **0.404** | 0.397 | 0.397 |

## C  DESIGN DETAILS OF MERINO

### C.1  SEARCH SPACE

Table 6 presents details of the search space defined for our entropy-driven design method. In addition, we set the embedding projection dimension as 768 and the maximum position embedding dimension as 2048. Our search space encapsulates over 216k different autoregressive transformer architectures.

Table 6: Search space hyperparameters for MeRino.

| | |
|---|---|
| Embedding Dimension - $E_i$ | [64, 128, 256, 384, 512, 640, 768, 896, 1024] |
| FFN Ratio - $R_i$ | [1, 1.5, 2, 2.5, 3, 3.5, 4] |
| Number of Layers Per Block - $L_i$ | [1, 2, 3, 4] |

### C.2  DETAIL STRUCTURE OF MERINO

The searched network structures of MeRino are listed in Tables 7. We use four blocks for our entropy-driven design. $E_i$ denotes the embedding dimension for each transformer block, $R_i$ denotes the FFN ratio, and $L_i$ denotes the number of layers (depth) of each transformer block.

Table 7: Structure Configuration of MeRino.

| Model | $E_i$ | $R_i$ | $L_i$ | Params | FLOPs |
|---|---|---|---|---|---|
| | [512, 512, 640, 896] | [1, 1, 1, 1] | [2, 3, 2, 1] | 52 M | 60 G |
| MeRino | [640, 768, 896, 1024] | [1, 1.5, 1, 1] | [2, 2, 2, 2] | 61 M | 110 G |
| | [640, 896, 1024, 1024] | [1.5, 1.5, 1, 1] | [3, 3, 2, 3] | 64 M | 160 G |

## C.3 EVOLUTIONARY ALGORITHM

We give a detailed description of the Evolutionary Algorithm (EA) and Mutation algorithm in Algorithm 1 and Algorithm 2, respectively.

---

**Algorithm 1** Evolutionary Algorithm

---

**Require:** Search space $\mathcal{D}$, number of iterations $T$, computation budget constraint $\mathcal{C}$, population size $M$, parent size $K$
**Ensure:** Optimal architecture $\mathcal{A}^*$
  Initialize population $\mathcal{P}$
  **while** $i \leq T$ **do**
    **while** $len(\mathcal{P}) < M$ **do**
      Random select $\mathcal{A}_i \in \mathcal{P}$ as parent.
      Mutate $\hat{\mathcal{A}}_i = \textbf{MUTATE}(\mathcal{A}_i, \mathcal{D})$
      **if** ComputeCost$(\hat{\mathcal{A}}_i) \leq \mathcal{C}$ **then**
        Calculate entropy $\mathcal{Z} = H(\hat{\mathcal{A}}_i)$
        Add $\hat{\mathcal{A}}_i$ to $\mathcal{P}$
      **else**
        Do nothing
      **end if**
    **end while**
    Remove $(M - K)$ networks with smallest entropy scores
  **end while**
  Return $\mathcal{A}^*$, the architecture with highest entropy in $\mathcal{P}$

---

**Algorithm 2** MUTATE

---

**Require:** Search space $\mathcal{D}$, architecture $\mathcal{A}_i$.
**Ensure:** Mutated architecture $\hat{\mathcal{A}}_i$
  Randomly select a block in $\mathcal{A}_i$
  Randomly alternate block depth, embedding dimension, and FFN ratio within a certain range
  Return the mutated architecture $\hat{\mathcal{A}}_i$

---

## D  LIMITATIONS

As no research is perfect, MeRino has several limitations as well. First, the design of MeRino explores entropy only from parameter subspace due to its straightforwardness. Further exploration of entropy in the feature space could provide a better theoretical understanding of transformer architecture and potentially lead to improved model designs. Second, our design only focuses on the "macro-structure" of the LLMs (channels/depths/heads). Other key components, such as residual connections, layer normalization, and nonlinear activations, are also essential to achieve good performance. However, the theoretical foundation for these components is not well-studied, especially from an information theory perspective. How to integrate these components in our entropy-based framework remains an open question and we would leav it for our future research.

