# OpenReview forum: "MeRino: Entropy-driven Design for Mobile-friendly Generative Language Models"
_ICLR.cc/2024/Conference — ICLR 2024 Conference Withdrawn Submission_

### Official Review · Reviewer_SWdN · 2023-10-28

**Soundness:** 2 fair
**Presentation:** 2 fair
**Contribution:** 2 fair
**Rating:** 3
**Confidence:** 4

**Summary:**

The paper investigates the efficient design of language models with the limit of computation resources. It proposes to optimize the entropy of transformer decoders with the extra consideration of decoder depth and takes a fast way to achieve it. Experiments are done on 12 zero-shot tasks, and compared with OPTs, Pythia, GPT-2, and Cerebras-GPT.

**Strengths:**

* The paper investigates an important and interesting problem, the efficient and expressive design of transformer language models.
* The proposed method is simple and does not cost much, which can be done on the CPU within minutes.

**Weaknesses:**

* In 3.2, the paper incorporates the depth constraints into the entropy-driven design, which looks empirical.
  * First, the paper thinks entropy can express expressiveness but the depth would hurt the effectiveness, and makes them a trade-off in Eq. 4. However, I think the expressiveness and effectiveness may include each other, which means the expressiveness of a model can indicate its effectiveness.
  * Second, the paper argues that the depth-width ratio can play an important role and give a hyper-parameter \beta to control it. However, I do not see much explanation about revising Eq. 2 to Eq. 4, and the choice of \beta value. For example, why choose 1/16 as the \beta value, and why make the ratio a multiplication factor in Eq. 4. More explanation is preferred.

* The experiments in this paper are not very solid.
  * First, the paper shall compare other methods for efficient model design like classic zero-shot NAS techniques in the main table. I see three figures about their training, but I think the concrete accuracy performance across tasks should be given in the main table. Also, the paper can compare with some classic zero-shot NAS techniques, which have been studied much in computer vision tasks. I understand the authors’ claim that they also need a little GPU support while this method does not. However, as this method needs large-scale pre-training, I do not think the little cost of zero-shot NAS really matters. Thus, the paper needs a more detailed comparison with other techniques to illustrate the superiority.
  * Second, the paper only works on very small models, which can not make me convinced of the effectiveness of the proposed method. As you can think, there are many techniques to largely affect the performance of small models. I understand that pre-training can cost much, but maybe the authors can operate on some pre-trained models, apply the search method, and do a little training, which will not cost much.
* Can the authors give some concrete examples of the searched architectures, which might help us better understand the method design and motivate others?

**Questions:**

Please check the weakness part.

---

### Official Review · Reviewer_BhDx · 2023-10-30

**Soundness:** 3 good
**Presentation:** 3 good
**Contribution:** 4 excellent
**Rating:** 3
**Confidence:** 4

**Summary:**

This work develop a Neural Architecture Search method for transformers to be deployed on mobile devices. The method estimates the entropy of the initial transformer architecture and the maximum entropy architecture is search with an evolutionary algorithm. Once selected the architecture is pretrained and compared against other models on zero-shot tasks.

Recommendation: I will start with a relatively low score tuntil the authors convince me that I can trust their evaluation. I am happy to increase my score if the authors explain why their experimental setup is comparable to the baselines in their work. I raise my score further if the authors provide a scaling analysis of larger models, or justify why such small models are interesting compared to models that are closer to 2 GB in size.

**Strengths:**

- strong correlations (0.86) between entropy measure and perplexity for many models trained from scratch compared to 0.77 for other methods
- extensive evaluation and ablations
- novel entropy approach for NAS

**Weaknesses:**

- Experimental setups not comparable. OPT uses less tokens; Pythia (and Cerebras-GPT?) uses a batch size of 1024 sequences for faster training (at the loss of performance); Merino uses a batch sizes of 512 (slower training, but better performance)
- The models are relatively small even for mobile devices. Since one could take a relatively large model, quantize it, and put it on a mobile device, this raises the question what the scaling behavior of the entropy method is. Does it scale to 1.3B? To 2.7B? I think any model under 2 GB is still pretty manageable for modern mobile devices and performance for such models is relevant.

**Questions:**

- How can FLOPs differ so much when the parameters are similar? Does it mean parameter sharing can be on/off for individual transformer blocks?
- How much does the architecture change for a particular FLOPs target if you draw a new random seed (different initial weights) for the optimization procedure?
- It seems you used a smaller batch size for training than Pythia did. Do you think that the evaluations with Pythia are still valid?
- Why did you optimize for those particular FLOPs and parameter counts?

---

### Official Review · Reviewer_e1q5 · 2023-10-30

**Soundness:** 2 fair
**Presentation:** 3 good
**Contribution:** 2 fair
**Rating:** 5
**Confidence:** 4

**Summary:**

The paper proposed a way to find a small size model which supposedly when trained from scratch can achieve similar performance as full-size model. This attribute is particularly useful for obtaining models from small devices.

**Strengths:**

1. Well written
2. The research direction is interesting.

**Weaknesses:**

1. The motivating theory sounds a bit weird to me.
2. Experiments are not convincing.

**Questions:**

1. The paper keeps saying it's a data-independent method, which sounds odds to me as usually entropy is related to the certain distribution so it shouldn't be data-independent. But indeed their entropy definition (They seem to quote from previous works which I didn't check the correctness) is data-independent. That makes me confused. My question is, shouldn't a model's generalization capability depends on the difficulty of the task? It simply cannot be data-independent.

In particular, if I have a dataset which is nothing but a random permutation of certain corpus. And then I split it into train/test so in this case the train/test distribution are the same but the task itself is impossible. In the motivating theory, the model should still generalize. Without a reasonable clarification on this, I think the paper should be rejected.

2. Since the model size is not large, I believe the paper should point out what's the performance difference between the approximation and the optimal solution. In this scale, the computational cost is not that high and we'd like to see the performance difference between approximated ones and non-approximated ones. That is,

2-a: What if I do full SVD instead of approximated one, the performance difference?
2-b: What if I solve the easy optimization problem by brute-force method on certain fine-grained grids, what's the performance difference?


3. I don't know why when comparing to the NAS approaches, it uses 1b dataset. LM task itself as a judgement doesn't make sense to me. I'd like to see the essentially tasks from Table 2 compared when evaluating NAS.

---

### Official Review · Reviewer_iksc · 2023-11-09

**Soundness:** 4 excellent
**Presentation:** 4 excellent
**Contribution:** 3 good
**Rating:** 6
**Confidence:** 3

**Summary:**

The authors propose an entropy-driven framework to achieve designing the efficient generative language models in resource constraint scenarios. Specifically, the authors leverage entropy to measure the effectiveness of deep neural networks based on previous methods, with consideration in the hardness to train a over-deep networks. An optimal efficient model designing strategy could be achieved by the proposed search process.

**Strengths:**

1. The proposed method achieve to find a solution for efficient transformers design, which solves an important topic.
2. The paper is well written, by clearly introducing the background information and the novel ideas.

**Weaknesses:**

1. The main concern I have is the experiment scale is too small. The larget model the proposed method used has 64M parameters, and the largest pre-trained model for comparison is OPT-350m. It's hard to make any conclusion when the overall performance is low. It would be better if the authors could provide the result on larger scale experiments.
2. I'm curious about the search result and some conclusion. It would be better if authors could provide some insights about the general rule for efficient model architecture design. (For example, perhaps deeper layers should have wider network designs.)

**Questions:**

See weakness.